# Destriping of Remote Sensing Images by an Optimized Variational Model

**DOI:** 10.3390/s23177529

**Published:** 2023-08-30

**Authors:** Fei Yan, Siyuan Wu, Qiong Zhang, Yunqing Liu, Haonan Sun

**Affiliations:** 1School of Electronic Information Engineering, Changchun University of Science and Technology, Changchun 130022, China; yanf@cust.edu.cn (F.Y.); zz_wusiyuan@163.com (S.W.); mzliuyunqing@163.com (Y.L.); 10yearsss2doublem@gmail.com (H.S.); 2Jilin Provincial Science and Technology Innovation Center of Intelligent Perception and Information Processing, Changchun 130022, China

**Keywords:** stripe noise removal, remote sensing images, *ℓ_p_* quasinorm, variational model, fast ADMM

## Abstract

Satellite sensors often capture remote sensing images that contain various types of stripe noise. The presence of these stripes significantly reduces the quality of the remote images and severely affects their subsequent applications in other fields. Despite the existence of many stripe noise removal methods in the research, they often result in the loss of fine details during the destriping process, and some methods even generate artifacts. In this paper, we proposed a new unidirectional variational model to remove horizontal stripe noise. The proposed model fully considered the directional characteristics and structural sparsity of the stripe noise, as well as the prior features of the underlying image, to design different sparse constraints, and the $\ell_{p}$ quasinorm was introduced in these constraints to better describe these sparse characteristics, thus achieving a more excellent destriping effect. Moreover, we employed the fast alternating direction method of multipliers (ADMM) to solve the proposed non-convex model. This significantly improved the efficiency and robustness of the proposed method. The qualitative and quantitative results from simulated and real data experiments confirm that our method outperforms existing destriping approaches in terms of stripe noise removal and preservation of image details.

## 1. Introduction

In recent years, remote sensing data have been widely applied in various fields. For instance, MODIS data include information on vegetation coverage, atmospheric conditions, and surface temperature, providing researchers with a vast amount of spectral data for Earth studies. However, it has been observed that certain bands in the level 1 and 2 MODIS data products publicly available on the official website of NASA exhibit prominent stripe noise. These stripe noise patterns can be broadly classified into two categories: periodic and non-periodic. Periodic stripes primarily arise from the need to stitch together data obtained from multiple detectors during the sensing process to acquire a sufficiently large focal plane, resulting in radiometric response discrepancies [1]. Additionally, the mechanical movements of the sensor itself introduce inevitable interferences and errors. Detector-to-detector and mirror-side stripes serve as two representative examples of periodic stripe noise [2]. On the other hand, non-periodic stripe noise manifests as random patterns, with uncertain lengths and occurrence positions. It is greatly influenced by spectral distribution and temperature factors, thus demanding higher post-processing requirements for the data. Apart from these factors, numerous other contributors to stripe noise in remote sensing imagery exist, making the effective removal of such noise a challenging task that researchers strive to address.

Hence, many scientific researchers have started participating in the task of removing stripe noise in order to improve the quality of remote sensing images. They have proposed numerous effective destriping methods.These destriping methods can be mainly categorized into the following groups: filter-based, statistical-based, model-optimization-based, and deep-learning-based methods. Filter-based methods primarily involve the use of filters of different sizes to eliminate stripe noise [3,4,5,6,7]. For example, in [3], Beat et al. proposed a filtering method for the removal of horizontal or vertical stripes based on a joint analysis of wavelets and Fourier transform. In [5], Cao et al. utilized wavelet decomposition to separate remote sensing images into different scales and applied one-dimensional guided filtering for destriping. Statistical-based methods focus on repairing images with stripe noise from the perspective of detector response and utilize the assumption of data similarity [8,9,10,11]. In [10], Carfantan et al. established a linear relationship through the gain estimation of detectors and proposed a statistical linear destriping (SLD) method for push-broom satellite imaging systems. Additionally, to remove irregular stripes from MODIS data, Shen et al. introduced a method based on local statistics and expected information [11]. In recent years, with the continuous development of deep learning methods in the field, this has been widely applied in various domains of image processing. Deep-learning-based destriping algorithms have also made progress in the research [12,13,14,15,16]. Chang et al. designed a HIS-DeNet network based on CNN to extract spectral and spatial information for removing stripe noise from hyperspectral images [13]. Huang et al. proposed a dual-network fusion denoising convolutional neural network ($D^{3}\mathrm{NNs}$) that could simultaneously remove both random noise and stripe noise [16].

Model-optimization-based methods are widely regarded as highly effective approaches at present. These methods mainly utilize the prior features and texture features of images to construct energy functionals with different regularization terms and obtain the final clean image by solving it mathematically. In 2010, Bouali et al. proposed the unidirectional variational method (UTV), which utilized the directional characteristics of stripe noise [2]. This method achieved significant results in the field of stripe noise removal for remote sensing images; however, it was prone to losing some fine details. Consequently, researchers started exploring the sparsity [17] and low rank [18] of the image itself and began addressing the limitations of this method from a mathematical perspective. As a result, many improved variants of UTV were studied. In [19], Zhou et al. proposed an adaptive coefficient unidirectional variational optimization model by replacing the $\ell_{2}$ norm in the regularization term with the $\ell_{1}$ norm. In [20], Liu et al. utilized the $l_{0}$ norm to characterize the global sparsity characteristics of stripe noise, effectively separating the stripe noise and achieving better stripe noise removal results. Chang et al. introduced the low-rank and sparse image decomposition (LRSID) model based on the low rank of a single image, extending the algorithm from 2D images to 3D hyperspectral images [21]. Compared to the original UTV method, the improved algorithms demonstrated a significantly enhanced destriping performance.

In summary, destriping methods in the research have been able to effectively remove stripe noise; however, there is still significant room for improvement in this area. Filtering-based and statistical-based methods have achieved good results in removing periodic stripes. However, they have often failed to deliver satisfactory results when dealing with complex non-periodic stripes. Deep-learning-based methods have high requirements for dataset preparation and choice of loss functions, and their applicability is limited to specific scenarios. In comparison, model-optimization-based methods have broader applicability; however, they still have their limitations. Many optimization models, for instance, consider the sparse characteristics of stripe noise structure. However, their exploration of these sparse characteristics is not sufficiently thorough. This deficiency leads to the loss of detailed information from the underlying image during the process of stripe removal. As a result, it is imperative to select appropriate norms that better characterize the relevant sparsity. Additionally, some optimization models involve overly intricate regularization constraints. Although they can yield improvements in destriping, the high number of parameters makes the process of parameter adjustment remarkably difficult. Hence, for more effective parameter configuration, the selection of simpler and more reasonable regularization constraints is of paramount importance.

In this paper, we propose a new destriping model based on $\ell_{p}$ quasinorm and unidirectional variation to overcome the limitations of previous methods. The model fully considers the prior characteristics of remote sensing images, as well as the directional and structural properties of stripe noise. We introduce the $\ell_{p}$ quasinorm to characterize the global sparsity of stripe noise and the local sparsity of the vertical image gradients. This norm facilitates the derivation of sparser solutions compared to the $\ell_{1}$ and $\ell_{2}$ norms, leading to superior stripe noise removal results. Additionally, the model employed the $\ell_{0}$ norm to capture the local sparsity of the horizontal gradient of stripe noise, further preventing the loss of image details during the destriping process. In the model-solving process, we adopted the fast alternating direction method of multipliers (ADMM) algorithm [22], which transforms the complex non-convex problem into simple subproblems for solution. The fast ADMM algorithm converges more rapidly and has shorter computation time compared to the traditional ADMM algorithm. The overall framework of the algorithm is presented in Figure 1. Finally, we conducted extensive experiments and compared our proposed method with six classical methods. The proposed method achieves superior stripe noise removal results and demonstrates a certain robustness. The contributions and innovations of this work can be summarized as follows:(1)We utilize the gradient information obtained from remote sensing image decomposition to design regularization constraints in different directions, effectively avoiding the ripple effect during the destriping process.(2)The $\ell_{p}$ quasinorm is introduced into the proposed model to better capture the relevant sparsity properties, thereby preserving a greater amount of fine details in the underlying image.(3)The fast ADMM algorithm is employed to solve the destriping model. It reduces the computational time, enabling efficient processing of large-scale data.

The subsequent sections are organized as follows: in Section 2, we introduce the relevant knowledge and research related to the proposed method. Section 3 provides a detailed explanation of the proposed stripe noise removal model and its solution. In Section 4, we conduct extensive experiments and compare our proposed method with six different approaches. The experimental results are analyzed and discussed in Section 5. Finally, Section 6 presents the conclusion of this study.

## 2. Related Work

### 2.1. Characteristics of Stripe Noise and UTV Model

It is commonly believed in the literature that the stripe noise present in remote sensing images is an additive noise. Assuming a remote sensing image $g\in L^{2}\left(\Omega \right)$, its degradation model can be expressed as
(1)g(x,y)=u(x,y)+n(x,y)
where g(x,y) represents the actual observed data obtained from the sensor, u(x,y) denotes the underlying image, and n(x,y) denotes the stripe noise. Based on this characteristic, many researchers in the field of stripe noise removal have focused their attention on variational model optimization methods.

The classical total variation (TV) model was initially proposed by Rudin et al. [23]. It plays a significant role in the field of image restoration. The model expression is as follows:(2)$$E\left(u\right)=\frac{1}{2}\int_{\Omega }{∥u-g∥}_{F}^{2}+\lambda \int_{\Omega }\sqrt{{\left(\frac{\partial u}{\partial x}\right)}^{2}+{\left(\frac{\partial u}{\partial y}\right)}^{2}}$$
in which λ is the coefficient of the regularization term, and its value plays a crucial role in the effectiveness of destriping methods. However, this model does not consider the correlated characteristics of stripe noise, resulting in limited effectiveness in removing stripe noise. In general, stripe noise tends to appear along the same direction. In this study, we focus on investigating horizontal stripe noise. To better analyze relevant prior characteristics, we extracted the image from the 27th band of MODIS data and computed the gradient information in different directions, as shown in Figure 2.

It can be observed that the stripe noise mainly affects the gradients in the vertical stripe direction, while the gradients along the stripe direction are minimally affected by the stripe noise. This implies that the horizontal gradient related to the stripe noise is much larger than the vertical gradient. Based on these observations, we can draw the following conclusions: (3)$$\left|\frac{\partial n(x,y)}{\partial x}\right|\ll \left|\frac{\partial n(x,y)}{\partial y}\right|$$

In this regard, some researchers have improved the total variation model by utilizing the directional characteristics of stripe noise and proposed the unidirectional total variation model. The mathematical formulation of the UTV model can be represented as
(4)$$E\left(u\right)=\int_{\Omega }{∥\frac{\partial (u-g)}{\partial x}∥}_{\begin{array}{c} 2 \end{array}}+\lambda {∥\frac{\partial u}{\partial y}∥}_{\begin{array}{c} 2 \end{array}}dxdy$$

The UTV model can be solved using the Euler–Lagrange equation. Furthermore, it is highly effective in removing stripe noise from remote sensing images. However, it only considers the local sparsity properties of gradients in different directions and overlooks the global sparsity properties of stripe noise. This can lead to the occurrence of ripple artifacts in the destriping process for remote sensing images heavily contaminated by stripe noise. Additionally, sparse constraints based on the $l_{2}$ norm can lead to the loss of details in striped images.

### 2.2. The Sparsity Analysis of the $l_{p}$ Quasinorm

Selecting an appropriate norm to characterize sparsity is crucial, as it better emphasizes the structural sparsity and low-rank properties of image and stripe components. In the field of image processing, the $l_{p}$ norm is typically defined as ${∥G∥}_{p}={\left(\sum_{i=1}^{N}\sum_{j=1}^{N}{\left|G_{ij}\right|}^{p}\right)}^{1/p}$, while the $l_{p}$ quasinorm is defined as ${∥G∥}_{p}^{p}=\sum_{i=1}^{N}\sum_{j=1}^{N}{\left|G_{ij}\right|}^{p}$. The value of p chosen for the $l_{p}$ quasinorm in this paper was 0<p<1. Compared to the $l_{2}$ norm and $l_{1}$ norm, the $l_{p}$ quasinorm offers greater flexibility and degrees of freedom [24]. In Figure 3, taking the gradient information in the vertical direction of the underlying image u as an example, it demonstrates the sparsity characterization capabilities of different norms. We can observe that the contours of the $l_{p}$ quasinorm are more likely to approach the coordinate axes, thereby inducing sparser solutions.

In the recent years, some researchers have achieved substantial progress in the field of image restoration by employing the $l_{p}$ quasinorm [25,26]. Therefore, in order to better describe the correlated sparsity characteristics between underlying image and stripe noise, we introduced the $l_{p}$ quasinorm into the destriping method in our study. The goal is to effectively remove stripes while preserving as much detail information in remote sensing images as possible.

## 3. Proposed Method

### 3.1. The Proposed Model

#### 3.1.1. Global Sparsity Constraint

Based on the previous analysis of stripe noise characteristics, we observed that stripe noise in remote sensing images typically appears as entire rows or columns. In practical scenarios, variations in length and the random positioning of the stripes can also exist. However, overall, stripe noise is sparse due to factors such as sensor offset. In many destriping algorithms [27,28,29,30,31], they commonly utilize the $l_{1}$ norm to characterize the sparse nature of the stripe noise. Considering the superior sparse representation capability of the $l_{p}$ quasinorm, we proposed a new global sparse representation:(5)$$R_{s}={∥u-g∥}_{p}^{p}$$

Even in the case of severe stripe noise pollution, Equation (Equation 5) exhibits a certain level of robustness. It ensures the similarity between denoised and original images, thereby minimizing the loss of excessive detail information.

#### 3.1.2. Local Sparsity Constraint

As shown in Figure 3, it can be observed that the gradient properties along the stripe direction and vertical direction exhibit different characteristics. Therefore, we need to establish separate sparse representation terms for each directional gradient. Firstly, from Figure 3b, we can observe that the gradient along the direction of the stripes in the remote sensing image with stripe noise is less affected by interference. This suggests that in addition to the inherent sparse distribution of the stripe noise, its horizontal gradient should also possess a certain level of sparsity. Furthermore, in [32], the researchers extracted features of the stripe noise using singular value decomposition and observed that the stripe noise exhibited a certain degree of low-rank property. Therefore, the gradient matrix of stripe noise along the horizontal direction consists mostly of zero elements. This conclusion remains valid regardless of the density and location of the stripe noise. Considering that the $l_{0}$ norm is capable of discerning zero and non-zero elements within a matrix, the regularization term for the horizontal direction can be expressed as:(6)$$R_{h}={∥\nabla_{x}\left(u-g\right)∥}_{0}$$

Furthermore, considering the prior information of the image itself, the stripe noise disrupts the local continuity of the underlying image, resulting in significant variations in the vertical gradients. To ensure the local continuity of the underlying image u, it is desirable to minimize the variations in $\nabla_{y}u$. This also indicates that the gradient along the vertical direction of the image should possess a certain degree of sparsity. Introducing the $l_{p}$ quasinorm, the sparse representation term for the vertical direction can be represented as:(7)$$R_{v}={∥\nabla_{y}u∥}_{p}^{p}$$

Finally, considering the stripe noise and the intrinsic characteristics of the underlying image, in combination with the aforementioned three sparse constraints, we propose a new model for remote sensing image destriping:(8)$$u=\arg\min_{u}{∥u-g∥}_{p}^{p}+\lambda_{1}{∥\nabla_{x}\left(u-g\right)∥}_{0}+\lambda_{2}{∥\nabla_{y}u∥}_{p}^{p}$$
where ${∥u-g∥}_{p}^{p}$ is referred to as the fidelity term, while ${∥\nabla_{x}\left(u-g\right)∥}_{0}$ and ${∥\nabla_{y}u∥}_{p}^{p}$ are referred to as the regularization terms. $\lambda_{1}$ and $\lambda_{2}$ represent the regularization coefficients, which determine the weighting values between the fidelity and regularization terms.

### 3.2. The Solution Based on Fast ADMM

Due to the non-convex and non-differentiable nature of the proposed model, some classical optimization algorithms may not be suitable for solving it. Therefore, the ADMM algorithm is commonly employed to solve such problems. Distinguished from other methods, this paper utilized the fast ADMM algorithm to solve the proposed model. The convergence speed of the original ADMM algorithm is O(1/k), while the fast ADMM algorithm can increase it to $O(1/k^{2})$ [22]. This significantly reduces the overall computation time of the algorithm, making it more efficient for handling large-scale datasets. The main idea is to introduce three intermediate variables, $M_{1}$, $M_{2}$, and $M_{3}$, to transform the unconstrained extremum problem presented in Equation (Equation 8) into a constrained extremum problem.
(9)$$\begin{array}{cc} \; & u=\arg\min_{u,M_{1},M_{2},M_{3}}{∥M_{1}∥}_{p}^{p}+\begin{array}{c} \lambda_{1} \end{array}{∥M_{2}∥}_{0}+\begin{array}{c} \lambda_{2} \end{array}{∥M_{3}∥}_{p}^{p}\\ & s.t.\;M_{1}=u-g,M_{2}=\nabla_{x}\left(u-g\right),M_{3}=\nabla_{y}u \end{array}$$

Then, we can obtain the augmented Lagrange function of the problem as follows: (10)$$\begin{array}{cc} L & ={∥M_{1}∥}_{p}^{p}+\begin{array}{c} \lambda_{1} \end{array}{∥M_{2}∥}_{0}+\begin{array}{c} \lambda_{2} \end{array}{∥M_{3}∥}_{p}^{p}\\ & +\langle Q_{1},\left(u-g\right)-M_{1}\rangle +\frac{\alpha_{1}}{2}{∥M_{1}-\left(u-g\right)∥}_{2}^{2}\\ & +\langle Q_{2},\nabla_{x}\left(u-g\right)-M_{2}\rangle +\frac{\alpha_{2}}{2}{∥M_{2}-\nabla_{x}\left(u-g\right)∥}_{2}^{2}\\ & +\langle Q_{3},\nabla_{y}u-M_{3}\rangle +\frac{\alpha_{3}}{2}{∥M_{3}-\nabla_{y}u∥}_{2}^{2} \end{array}$$
where $Q_{1}$, $Q_{2}$ and $Q_{3}$ are the Lagrange multipliers. $\alpha_{1}$, $\alpha_{2}$ and $\alpha_{3}$ are the coefficients of the penalty terms. We needed to solve the subproblems for each variable individually. Additionally, auxiliary variables $M_{1a}$, $M_{2a}$, $M_{3a}$, $Q_{1a}$, $Q_{2a}$, $Q_{1a}$, and $\eta_{i}$ were introduced to accelerate the iteration process. Among them, $M_{1a}$ represents the intermediate variable in the iteration of $M_{1}$ and is equivalent to $M_{1}$ in solving the subproblem, and the other variables are similar.

(1)The subproblem related to u is


(11)
$$\begin{array}{cc} u & =\arg\min_{u}\langle Q_{1},\left(u-g\right)-M_{1}\rangle +\frac{\alpha_{1}}{2}{∥M_{1}-\left(u-g\right)∥}_{2}^{2}\\ & +\langle Q_{2},\nabla_{x}\left(u-g\right)-M_{2}\rangle +\frac{\alpha_{2}}{2}{∥M_{2}-\nabla_{x}\left(u-g\right)∥}_{2}^{2}\\ & +\langle \;Q_{3},\nabla_{y}u-M_{3}\rangle +\frac{\alpha_{3}}{2}{∥M_{3}-\nabla_{y}u∥}_{2}^{2} \end{array}$$


Considering the decoupling of other variables from u, Equation (Equation 11) can be represented using the convolution as follows: (12)$$\begin{array}{cc} u & =\arg\min_{u}\frac{\alpha_{1}}{2}{∥M_{1}-\left(u-g\right)-\begin{array}{c} \frac{Q_{1}}{\alpha_{1}} \end{array}∥}_{2}^{2}+\frac{\alpha_{2}}{2}{∥M_{2}-K_{x}*\left(u-g\right)-\begin{array}{c} \frac{Q_{2}}{\alpha_{2}} \end{array}∥}_{2}^{2}\\ & +\frac{\alpha_{3}}{2}{∥M_{3}-K_{y}*u-\begin{array}{c} \frac{Q_{3}}{\alpha_{3}} \end{array}∥}_{2}^{2} \end{array}$$
where ∗ represents the convolution operation, $K_{x}=\left[-1,1\right]$ represents the convolution kernel for horizontal differencing, and $K_{y}={\left[-1,1\right]}^{T}$ presents the convolution kernel for vertical differencing.

Then, Equation (Equation 12) can be solved using the convolution theorem and fast Fourier transform (FFT). After introducing acceleration auxiliary variables, the iterative formula for u can be obtained as follows: (13)$$u^{\left(k+1\right)}=F^{-1}\left(\frac{\varphi }{\alpha_{1}+\alpha_{2}{\left[F(K_{x})\right]}^{*}\circ F\left(K_{x}\right)+\alpha_{3}{\left[F(K_{y})\right]}^{*}\circ F\left(K_{y}\right)}\right)$$
in which
(14)$$\begin{array}{cc} \varphi & =\alpha_{1}\left(F\left(M_{1a}^{\left(k\right)}\right)-F\left(\begin{array}{c} \frac{Q_{1a}^{\left(k\right)}}{\alpha_{1}} \end{array}\right)\right)+\alpha_{2}{\left[F(K_{x})\right]}^{*}\circ \left(F\left(M_{2a}^{\left(k\right)}\right)-F\left(\begin{array}{c} \frac{Q_{2a}^{\left(k\right)}}{\alpha_{2}} \end{array}\right)\right)\\ & +\alpha_{3}{\left[F(K_{x})\right]}^{*}\circ \left(F\left(M_{3a}^{\left(k\right)}\right)-F\left(\begin{array}{c} \frac{Q_{3a}^{\left(k\right)}}{\alpha_{3}} \end{array}\right)\right)+\left(\alpha_{1}+\alpha_{2}{\left[F(K_{x})\right]}^{*}\circ F\left(K_{x}\right)\right)\circ F\left(g\right) \end{array}$$
where ∘ denotes component-wise multiplication, F denotes the operator for the Fourier transform, $F^{-1}$ denotes the operator for the inverse Fourier transform, and ${\left[F(K_{i})\right]}^{*}$ denotes the conjugate map for $F(K_{i})$.

(2)The subproblem related to $M_{1}$ is


(15)
$$M_{1}=\arg\min_{M_{1}}{∥M_{1}∥}_{p}^{p}+\langle Q_{1},\left(u-g\right)-M_{1}\rangle +\frac{\alpha_{1}}{2}{∥M_{1}-\left(u-g\right)∥}_{2}^{2}$$


Considering the decoupling of $M_{1}$ from the other variables, the subproblem for $M_{1}$ can be represented as follows: (16)$$M_{1}=\arg\min_{M_{1}}{∥M_{1}∥}_{p}^{p}+\frac{\alpha_{1}}{2}{∥M_{1}-\left(u-g\right)-\begin{array}{c} \frac{Q_{1}}{\alpha_{1}} \end{array}∥}_{2}^{2}$$

It can be solved using the shrinkage operator with soft thresholding [33,34], resulting in the accelerated iterative formula for $M_{1}$: (17)$$M_{1}^{\left(k+1\right)}=shrink_{p}\left(u^{\left(k+1\right)}-g+\begin{array}{c} \frac{Q_{1a}^{\left(k\right)}}{\alpha_{1}} \end{array},\frac{1}{\alpha_{1}}\right)$$
where
(18)$$shrink_{p}\left(m,n\right)=max\left\{\left|m\right|-n^{2-p}{\left|m\right|}^{p-1},0\right\}\cdot \frac{m}{\left|m\right|}$$

(3)The subproblem related to $M_{2}$ is


(19)
$$M_{2}=\arg\min_{M_{2}}\begin{array}{c} \lambda_{1}{∥M_{2}∥}_{0} \end{array}+\langle Q_{2},\nabla_{x}\left(u-g\right)-M_{2}\rangle +\frac{\alpha_{2}}{2}{∥M_{2}-\nabla_{x}\left(u-g\right)∥}_{2}^{2}$$


Similarly, due to the decoupling of $M_{2}$ from the other variables, we can obtain the following expression: (20)$$M_{2}=\arg\min_{M_{2}}\lambda_{1}{∥M_{2}∥}_{0}+\frac{\alpha_{2}}{2}{∥M_{2}-\nabla_{x}\left(u-g\right)-\frac{Q_{2}}{\alpha_{2}}∥}_{2}^{2}$$

Then, according to the hard thresholding shrinkage theorem [35,36], $M_{2}$ can be updated in an accelerated manner: (21)$$M_{2}^{\left(k+1\right)}=hard\left(\nabla_{x}\left(u^{\left(k+1\right)}-g\right)+\begin{array}{c} \frac{Q_{2a}^{\left(k\right)}}{\alpha_{2}} \end{array},\sqrt{\frac{2\lambda_{1}}{\alpha_{2}}}\right)$$
where
(22)$$hard\left(m,n\right)=\left\{\begin{array}{c} 0,\;\;\left|m\right|<n\\ m,\;\;\left|m\right|\geq n \end{array}\right.$$

(4)The subproblem related to $M_{3}$ is


(23)
$$M_{3}=\arg\min_{M_{3}}\lambda_{2}\begin{array}{c} {∥M_{3}∥}_{p}^{p} \end{array}+\langle Q_{3},\nabla_{y}u-M_{3}\rangle +\frac{\alpha_{3}}{2}{∥M_{3}-\nabla_{y}u∥}_{2}^{2}$$


In a similar manner to $M_{1}$, we can derive the accelerated iterative formula for $M_{3}$ as follows: (24)$$M_{3}^{\left(k+1\right)}=shrink_{p}\left(\nabla_{y}u^{\left(k+1\right)}+\begin{array}{c} \frac{Q_{3a}^{\left(k\right)}}{\alpha_{3}} \end{array},\frac{\lambda_{2}}{\alpha_{3}}\right)$$

Finally, utilizing the gradient ascent method, the Lagrange multipliers $Q_{1}$, $Q_{2}$, and $Q_{3}$ can be updated by
(25)$$Q_{1}^{\left(k+1\right)}=Q_{1a}^{\left(\begin{array}{c} k \end{array}\right)}+\alpha_{1}\left(u^{\left(k+1\right)}-g-M_{1}^{\left(k+1\right)}\right)$$
(26)$$Q_{2}^{\left(k+1\right)}=Q_{2a}^{\left(\begin{array}{c} k \end{array}\right)}+\alpha_{2}\left(\nabla_{x}\left(u^{\left(k+1\right)}-g\right)-M_{2}^{\left(k+1\right)}\right)$$
(27)$$Q_{3}^{\left(k+1\right)}=Q_{3a}^{\left(\begin{array}{c} k \end{array}\right)}+\alpha_{3}\left(\nabla_{y}u^{\left(k+1\right)}-M_{3}^{\left(k+1\right)}\right)$$

Following the determination of the iterative formulas for each variable, the accelerated iteration process can be initiated. The calculation formula for the primal-dual residual, which is crucial for accelerated iterations, is as follows: (28)$$Z_{i}^{\left(k\right)}=\alpha_{i}^{-1}∥Q_{i}^{\left(k\right)}-Q_{ia}^{\left(k\right)}∥+\alpha_{i}∥M_{i}^{\left(k\right)}-M_{ia}^{\left(k\right)}∥$$

If condition $Z_{i}^{\left(k+1\right)}<\mu Z_{i}^{\left(k\right)}$ is satisfied, then update the iteration step length. Otherwise, the algorithm is restarted using the result of the previous iteration as the initial value. The auxiliary variable can be updated by the following expressions: (29)$$\eta_{i}^{\left(k+1\right)}=\frac{1+\sqrt{1+4{\left(\eta_{i}^{\left(k\right)}\right)}^{2}}}{2}$$
(30)$$M_{ia}^{\left(k+1\right)}=M_{i}^{\left(k+1\right)}+\frac{\eta_{i}^{\left(k\right)}-1}{\eta_{i}^{\left(k+1\right)}}\left(M_{i}^{\left(k+1\right)}-M_{i}^{\left(k\right)}\right)$$
(31)$$Q_{ia}^{\left(k+1\right)}=Q_{i}^{\left(k+1\right)}+\frac{\eta_{i}^{\left(k\right)}-1}{\eta_{i}^{\left(k+1\right)}}\left(Q_{i}^{\left(k+1\right)}-Q_{i}^{\left(k\right)}\right)$$
where i=1,2,3, and μ is a scaling coefficient that approximates 1.

Thus far, with all subproblems solved, the overall algorithm can be summarized as presented in Algthorm 1.
**Algorithm 1:** The proposed destriping model with Fast ADMM**Input:** Degraded image g and related parameter $\lambda_{1}$, $\lambda_{2}$, $\alpha_{1}$, $\alpha_{2}$, and $\alpha_{3}$1: **Initialize:** Set $u^{\left(0\right)}=M_{i}^{\left(0\right)}=M_{ia}^{\left(0\right)}=Q_{i}^{\left(0\right)}=Q_{ia}^{\left(0\right)}=0$ (i=1,2,3), $\eta_{i}^{\left(0\right)}=1$, $\varepsilon =10^{-4}$, $n_{max}=200$.2: While: $\frac{{∥u^{\left(k+1\right)}-u^{\left(k\right)}∥}_{2}^{2}}{{∥u^{\left(k\right)}∥}_{2}^{2}}<\varepsilon$ and $n<n_{max}$3: update $u^{\left(k+1\right)}$ by using Equation (Equation 13)4: update $M_{1}^{\left(k+1\right)}$, $M_{2}^{\left(k+1\right)}$, and $M_{3}^{\left(k+1\right)}$ by using Equations (Equation 17), (Equation 21) and (Equation 24)5: update $Q_{1}^{\left(k+1\right)}$, $Q_{2}^{\left(k+1\right)}$, and $Q_{3}^{\left(k+1\right)}$ by using Equations (Equation 25)–(Equation 27)6: update $Z_{i}^{\left(k+1\right)}$ by using Equation (Equation 28)7: **if** $Z_{i}^{\left(k+1\right)}<\mu Z_{i}^{\left(k\right)}$, i=1,2,3 then8: update $\eta_{i}^{\left(k+1\right)}$, $M_{ia}^{\left(k+1\right)}$ and $Q_{ia}^{\left(k+1\right)}$ by using Equations (Equation 29)–(Equation 31)9: **else**10: $\eta_{i}^{\left(k+1\right)}=1$, $M_{ia}^{\left(k+1\right)}=M_{i}^{\left(k+1\right)}$, $Q_{ia}^{\left(k+1\right)}=Q_{i}^{\left(k+1\right)}$, $Z_{i}^{\left(k+1\right)}=\mu^{-1}Z_{i}^{\left(k\right)}$, i=1,2,311: **end if**12: n=n+113: **End While****Output:** Destriped image u

## 4. Experiment Results

To validate the effectiveness and generalizability of the proposed method, we conducted separate tests on simulated and real stripe noise. The experiments involved a significant number of comparative experiments to evaluate the performance of the proposed method on different types of stripe noise, ensuring the logical rigor of the experiments. For the comparative experiments, we analyzed and compared six typical destriping methods. Among them, WAFT [3] and WLS [37] are filtering-based and statistical-based methods. UTV [2] is a model optimization method based on the $l_{2}$ norm. SAUTV [19] is an earlier model optimization algorithm based on the $l_{1}$ norm, while GSLV [27] and RBSUTV [38] are more advanced model optimization methods proposed in the research in recent years. Furthermore, all our experiments were conducted on a personal computer with an Intel(R) Core (TM) i7-6700 CPU @ 3.40 GHz and 16 GB RAM, using MATLAB R2022b.

In the evaluation of the method, this study employed a comprehensive evaluation approach that combined subjective and objective assessments, resulting in the more persuasive evaluation results. In terms of subjective evaluation, the effectiveness of different methods in removing stripes can be directly observed by examining the restored remote sensing images and their corresponding stripe noise maps. We also zoomed in on some areas with noticeable differences and marked them with red boxes in the images. Furthermore, we plotted the mean cross-track profiles and mean column power spectrum of the restored remote sensing images to better demonstrate the differences in the stripe noise removal results among the different methods. In terms of objective evaluation, we use different reference indicators for simulated data and real data. For the simulated data, since the original images are available, we used peak-signal-to-noise ratio (PSNR) and structural similarity (SSIM) [39] as reference metrics. For the real data, we employed two commonly used no-reference evaluation metrics in the field of stripe noise removal: mean relative deviation (MRD) [40,41] and inverse coefficient of variation (ICV) [20,42].

### 4.1. Simulated Data Experiments

In this section, we conducted extensive simulation experiments to validate the superiority of the algorithm. In Figure 4, we selected six remote sensing images captured by different sensors for conducting simulated stripe noise experiments. Among them, Figure 4a,b,f are MODIS data products obtained from 1B-level calibrated radiance, which can be downloaded from the official website of NASA [43]. Figure 4c is a hyperspectral image captured by the Tiangong-1 satellite. Figure 4d is a hyperspectral image of Washington DC Mall, which can be obtained from the relevant website [44]. Figure 4e is captured by the VIIRS sensor and is included in the “Earth at Night” collection, which can also be obtained from the official website of NASA [45].

In the experiment, the simulated stripe noise can be mainly classified into two categories: periodic and non-periodic. Inspired by the ideas presented in [31], we verified the effectiveness and robustness of the proposed method by adding stripes of different intensities and ratios. During destriping process, the remote sensing image was operated in MATLAB as an eight-bit encoded matrix. Therefore, the intensity of the stripe noise could be selected within the range of [0, 255]. The ratio refers to the ratio of the number of rows where stripe noise appears to the total number of rows in the remote sensing image matrix and can be selected within the range of [0, 1]. Additionally, to facilitate better comparison of the stripe noise removal effects of different methods, we normalized the stripe noise. Moreover, in the simulated experiment, we employed two important objective reference metrics: PSNR and SSIM. The calculation methods for these metrics are as follows: (32)$$\mathrm{PSNR}\left(I,R\right)=10\log\frac{255\times 255}{\frac{1}{mn}\sum_{i=1}^{m}\sum_{j=1}^{n}{\left(I_{ij}-R_{ij}\right)}^{2}}$$
where I represents the original image, R represents the destriped image, and mn represents the total number of pixels in the image.
(33)$$\mathrm{SSIM}\left(I,R\right)=\frac{\left(M_{I}M_{R}+{\left(255k_{1}\right)}^{2}\right)\left(2\sigma_{IR}+{\left(255k_{2}\right)}^{2}\right)}{\left(M_{I}^{2}+M_{R}^{2}+{\left(255k_{1}\right)}^{2}\right)\left(\sigma_{I}^{2}+\sigma_{R}^{2}+{\left(255k_{2}\right)}^{2}\right)}$$
where $M_{I}$ and $M_{R}$ respectively denote the pixel means of images I and R. $\sigma_{IR}$ denotes the covariance between g and u, while $\sigma_{I}^{2}$ and $\sigma_{R}^{2}$ respectively denote the variances of the images I and R. k1 and k2 are constants used in the calculations.

#### 4.1.1. Periodic Stripe Noise

For the assessment of periodic stripe noise, we selected three different remote sensing images and added stripe noise of varying intensities and ratios for experimentation. In Figure 5, through the restored remote sensing images, we observed that all the methods effectively removed the stripe noise. However, the WAFT method introduced some ripple artifacts in the denoised image, while UTV and SAUTV caused the blurring of the edge details in the recovered image. The other three methods and the proposed method performed well, with no significant visual differences observed.

In Figure 6, the denoising results of most methods are similar to Figure 5, except for the WLS method, which still exhibits some noticeable stripes in the denoised images. This was because we added stripe noise with different ratios to the two remote sensing images, resulting in different frequencies. The WLS method was unable to accurately compute the local linear relationship between the image and stripe noise through guided filtering, thus leading to the incomplete removal of stripe noise. This indicates that filter-based and statistical-based stripe noise removal algorithms are not universally applicable. In Figure 7, we present the estimates of stripe noise of Figure 6. For several model optimization algorithms, it can be observed that whether it is UTV based on the $l_{2}$ norm or SAUTV, GSLV, and RBSUTV based on the $l_{1}$ norm, a relatively large amount of underlying image information is removed while destriping. In contrast, the proposed method removed the least amount of underlying image detail information. It indicates that the $l_{p}$ quasinorm has a better capability to characterize the sparse nature of stripe noise compared to the $l_{1}$ and $l_{2}$ norms.

To distinguish the subtle differences between the different methods, we increased the ratio of periodic stripes in Figure 8 and plotted the corresponding metric curves. Figure 9 presents the mean cross-track profiles of the stripe removal results for each method. Although some methods showed no significant differences in the visual results of stripe removal compared to the proposed method, the mean cross-track profile of the proposed method was noticeably superior to these methods. It can be observed that, except for the proposed method, the mean cross-track profiles of other methods have a certain deviation from the original image at the peak. This also indicates that the $l_{1}$ and $l_{2}$ norms inadequately capture the sparse characteristics of stripe noise, resulting in the loss of detail information during the process of destriping. Figure 10 displays the mean column power spectrum for each method. The main frequencies of the stripe noise are concentrated around 0.1, 0.2, 0.3, and 0.4, which align with the actual situation. Around the frequency of 0.05, the curve of the proposed method closely fits the original curve compared to the other methods. This indicates that the proposed method outperforms the other methods in preserving details.

Furthermore, from the results presented in Table 1, it can be observed that WAFT and WLS exhibit a good performance in removing stripes at lower levels. However, as the level of stripe noise increases, their ability to remove the noise deteriorates rapidly. Consistently with the results presented in Figure 6 and Figure 7, UTV, SAUTV, and GSLV all result in the loss of some details from the underlying image when removing stripe noise, leading to lower PSNR and SSIM values compared to the proposed method. The RBSUTV method, due to its specific characteristics, performs better than the proposed method in handling low-level stripe noise; however, it shows a poor performance when dealing with high-level stripe noise. In comparison, the proposed method is superior to other methods in most cases and demonstrated robustness. Although it may not surpass RBSUTV in some situations, the difference between the two is minimal.

#### 4.1.2. Nonperiodic Stripe Noise

For experiments conducted on non-periodic stripe noise, we also selected three examples for demonstration and explanation, with the relevant results shown in Figure 11, Figure 12, Figure 13, Figure 14, Figure 15 and Figure 16. Among them, Figure 11, Figure 12 and Figure 14 show the results of removing different levels of non-periodic stripe noise. Overall, the results are similar to the handling of periodic stripe noise; however, there are still some notable points. In Figure 11, the horizontal streets are mistakenly treated as stripe noise and removed in the results of SAUTV and RBSUTV. This is because these edge structures are very similar to stripe noise and are prone to being misprocessed. Therefore, during the destriping process, the consideration of global sparsity is necessary. Additionally, due to the overall smoothness of the simulated remote sensing image presented in Figure 12, there is less detailed information of the underlying image contained in the presented stripe noise in Figure 13. This suggests that the image itself also has an impact on the stripe removal noise capability of the algorithm.

In Figure 15, it can be observed that, unlike periodic stripes, the mean cross-track profiles of non-periodic stripe noise appear more chaotic. Additionally, from the mean column power spectrum in Figure 16, it is evident that the frequencies of non-periodic stripe noise are not concentrated but rather dispersed. As a result, non-periodic stripe noise has a significant impact on filtering-based and statistical-based destriping methods. Furthermore, from the PSNR and SSIM values of the recovered images using different methods presented in Table 2, we can observe that the performance of WAFT and WLS noticeably decreases compared to removing periodic stripes, and other algorithms also show slight decreases. This indicates that the difficulty of destriping increases when non-periodic stripe noise is present. However, the overall trends remain consistent with periodic stripe noise. The proposed method in this paper still outperforms other algorithms in terms of performance.

### 4.2. Real Data Experiments

In the experiments conducted with the use of real data, we selected four remote sensing images from the MODIS dataset. Figure 17a was obtained from the 33rd band of the MODIS data, primarily affected by non-periodic stripe noise. Figure 18a and Figure 19a were both obtained from the 27th band of the MODIS data. Figure 18a is primarily contaminated by periodic stripes, while Figure 19a is primarily contaminated by non-periodic stripes. This indicates that the stripe noise in the same band is not fixed. Figure 20a was obtained from the 31st band of the MODIS data, and it exhibits a high ratio of stripe noise but with a relatively low intensity level.

Furthermore, since there are no original images available for the real data, we selected two commonly used no-reference metrics, MRD and ICV, to calculate the objective evaluation metrics. These metrics were used to quantitatively assess the performance of different methods, and their calculation methods are as follows: (34)$$\mathrm{MRD}=\frac{1}{mn}\sum_{i=1}^{m}\sum_{j=1}^{n}\frac{\left|X_{\mathrm{ij}}-Y_{\mathrm{ij}}\right|}{X_{\mathrm{ij}}}\times 100\%$$
where mn represents the number of pixels in the region unaffected by stripes, $X_{\mathrm{ij}}$ represents the pixel value of the original image, and $Y_{\mathrm{ij}}$ represents the pixel value of the image after removing the stripes. Considering the characteristics of stripe noise, we selected individual rows without stripe noise as the region unaffected by stripe noise to calculate the MRD index. Furthermore, we calculated the mean value from multiple calculations as the final result to avoid the randomness of the results.
(35)$$\mathrm{ICV}=\frac{Y_{m}}{Y_{std}}$$
where $Y_{m}$ represents the average value of pixels, and $Y_{std}$ represents the standard deviation of pixels. As suggested in [2], we selected a 10 × 10 pixel window to compute the ICV index in homogeneous regions of the destriping image. To better differentiate the differences between different algorithms, we performed multiple calculations in the experiment and selected the set of data with the highest value as the final result.

Figure 17, Figure 18, Figure 19 and Figure 20 depict the restored images after applying different methods for removing stripe noise in the real data. It is worth noting that Figure 17 presents extremely dark regions, which are highly challenging for destriping. As a result, most methods exhibited artifacts in that specific area. However, the proposed method effectively avoided the generation of artifacts by incorporating a vertical sparsity constraint with the $l_{p}$ quasinorm. Furthermore, Figure 19 reveals that, in practical scenarios, non-periodic stripe noise appears randomly in terms of location, length, and intensity, significantly increasing the difficulty of removing such stripes. As observed in the local magnification images, both WAFT and WLS still exhibit residual stripe noise in their restored images. Even the RBSUTV method, which demonstrates a good performance on destriping in simulated experiments, failed to achieve satisfactory results when confronted with this type of stripe. In this case, the proposed method still achieved excellent destriping results. This strongly indicates the superior capability of the $l_{p}$ quasinorm in characterizing the sparse nature of underlying image, enabling better removal of stripe information contained in the underlying image.

In Figure 21, we can observe that the mean cross-track profiles of the destriping results obtained from WAFT, UTV, and SAUTV align with the trend of the original image. However, their curves appear overly smooth, indicating a loss of detailed information. In Figure 21c, some prominent spikes are present, indicating the presence of residual stripe noise in the denoised result of WLS. The RBSUTV method shows significant deviations in certain row mean values compared to the original image, which could be attributed to the choice of regularization terms. In comparison, the curves of GSLV and the proposed method align most closely with the trend of the original image. Due to the relatively low intensity of the stripe noise, all methods effectively removed the stripe noise. Consequently, in Figure 22, there is minimal difference in the normalized power spectrum of the stripe removal results among the various methods.

Additionally, Table 3 presents the objective evaluation metrics, MRD and ICV, for Figure 17, Figure 18, Figure 19 and Figure 20, with the best results being highlighted. From the table, we can observe that the proposed method outperforms the other methods in most cases. Although it may not achieve the best results in some cases, the disparity arising from the optimal results remains minor. This further demonstrates the effectiveness and robustness of the proposed method.

## 5. Discussion

### 5.1. Discussion of Experiment Results

In Section 4, we conducted extensive simulations and real experiments. The results show that the proposed method outperforms other methods in most cases; however, there are still some details worth discussing. From the visual results of destriping, we observed that WAFT and WLS performed poorly as the restored images still contained some residual stripe noise. For the model optimization methods, both UTV and SAUTV tend to blur edge structures and even introduce artifacts during the stripe removal process. Although there is no significant visual difference in the stripe removal results between GSLV and RBSUTV, the loss of detail information can still be detected from the correlation curve graphs. It is evident that the characterization of sparse properties by $l_{1}$ or $l_{2}$ norms is limited. Therefore, introducing the $l_{p}$ quasinorm into the sparse constraint proves to be beneficial. Although the proposed algorithm achieved good overall results, there are still some gaps in capturing fine details compared to the original image. From Table 1 and Table 2, we can observe that the proposed method presents some differences compared to the other methods when dealing with low-intensity stripe noise. This indicates that our consideration of sparse constraints on the image component and stripe component may not be comprehensive enough. Additionally, in Table 3, there are differences between the ICV metric of the proposed method and some other methods. Part of the reason could be that the selected region pixels were not evenly distributed, leading to less accurate computation results. Therefore, when evaluating the performance of the algorithm, it is important to consider other metrics for a comprehensive assessment.

Computer performance is very powerful at present, and algorithm runtime is not the primary concern for most people. However, with the advent of the big data era, algorithm runtime has also become a noteworthy factor in the research. In [22], the fast ADMM algorithm was proposed to speed up the entire iteration process by adjusting the step size. Therefore, we adopted the fast ADMM algorithm in our study to solve the proposed optimization model, greatly reducing the overall runtime of the proposed method. In Table 4, the runtime for solving six different-sized images using both ADMM and Fast ADMM is presented. It can be observed that as the image size increases, the Fast ADMM algorithm gains a more pronounced advantage. Additionally, during the experimental process, we also found that the Fast ADMM algorithm sometimes contributes to slight improvements in experimental results.

### 5.2. Analysis of the Parameters

The regularization coefficients have a decisive impact on the destriping performance of the optimization model, and selecting suitable coefficients is crucial. In practical scenarios, it is challenging to design universal coefficients for different types of stripe noise, and they are often determined through extensive experimentation processes. The main parameters of the proposed method are the two regularization parameters: $\lambda_{1}$ and $\lambda_{2}$. Additionally, the choice of these parameters depends on the level of stripe noise. Based on the analysis of the proposed model and extensive experiments, the following conclusions can be drawn:(1)When the stripe noise is weak, it is generally recommended to select a larger value for $\lambda_{1}$, which increases the weight of the horizontal stripe component, better preserving the details of the underlying image.(2)When the stripe noise is strong, it is generally recommended to select a larger value for $\lambda_{2}$, which increases the weight of the vertical image component, enhancing the destriping capability.

However, the interaction between these two parameters should not be ignored. To obtain the optimal values for both parameters, it would require exhaustive testing in a two-dimensional space, which is a formidable task. Therefore, following the recommendation in [31], we employed a greedy algorithm combined with extensive experiments to rapidly determine parameter settings that yield better destriping effects. Specifically, the suggested range for $\lambda_{1}$ was [1, 100], and for $\lambda_{2}$ was [0.01, 1].

### 5.3. Limitation

The proposed method achieved a good destriping results in many cases; however, it still presented certain limitations. Remote sensing images obtained from sensors are often subject to various interferences, resulting in the presence of complex stripe noise in the images. The proposed method did not incorporate adaptive parameter selection. Therefore, selecting suitable parameters for different types of stripe noise became challenging and requires significant time and computational effort. Additionally, as shown in Figure 23a, some small-scale stripe noises disrupt the overall sparsity and low-rank properties of the stripe noise. Consequently, the proposed method was unable to completely remove such noise, and similar situations are observed in other methods [46] as well. In the experiments, we mitigated this limitation by adjusting the coefficients of the regularization terms; however, the overall image restoration quality was significantly compromised. Therefore, a research direction for future work would be to develop methods to effectively remove these small-scale random stripe noises while preserving the overall image restoration quality.

## 6. Conclusions

In this paper, we proposed a univariate variational model based on the $l_{p}$ quasinorm for removing stripe noise and applied it to different types of stripes in remote sensing data. The model considered the sparsity and low-rank properties of the stripe and image components separately. By introducing the $l_{p}$ quasinorm, it effectively avoided the loss of details during the destriping process. In the model-solving process, a fast ADMM algorithm was employed to speed up the convergence during iterations. Finally, extensive simulations and real experiments were conducted to compare the proposed method with six different methods. From both subjective and objective experimental results, it can be observed that the proposed method not only effectively removes stripes but also better preserves the details of the original image compared to the other methods. Even under severe stripe contamination, the proposed method achieved a good removal performance and demonstrated strong robustness. Furthermore, the fast ADMM algorithm reduces the running time of the proposed method, providing it with better prospects in the era of big data.

In addition, although this paper focused on the application of the proposed method to remote sensing data, its applicability was not limited to this specific domain. The method can be extended to the entire field of stripe noise removal. Therefore, in order to adapt to a wider range of application scenarios, we will address the limitations of the proposed method and make improvements in the future work. Additionally, we will continue to explore additional characteristics of the stripe and image components to better remove stripe noise. Moreover, with the continuous development and improvement of neural network methods, we will also consider integrating deep learning methods with traditional methods to obtain more effective approaches for destriping.

## Figures and Tables

**Figure 1 sensors-23-07529-f001:**
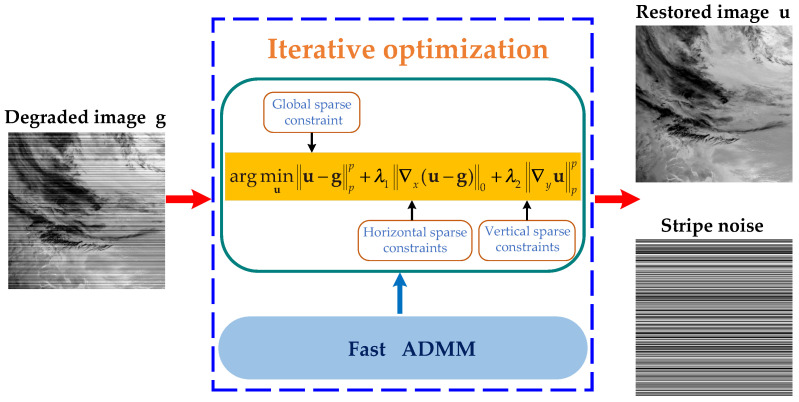
The schematic diagram of the proposed destriping method.

**Figure 2 sensors-23-07529-f002:**
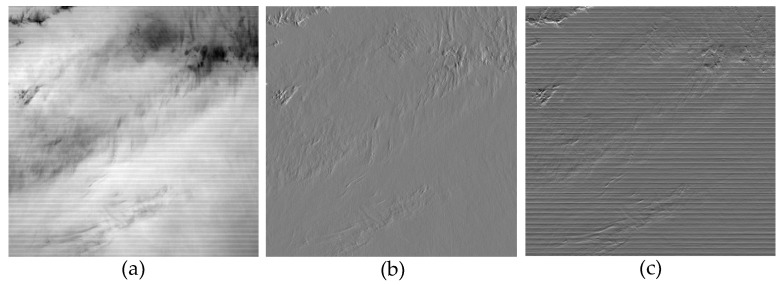
Gradient properties in different directions of MODIS band 27 (**a**) Original striped image. (**b**) Horizontal gradient property. (**c**) Vertical gradient property.

**Figure 3 sensors-23-07529-f003:**
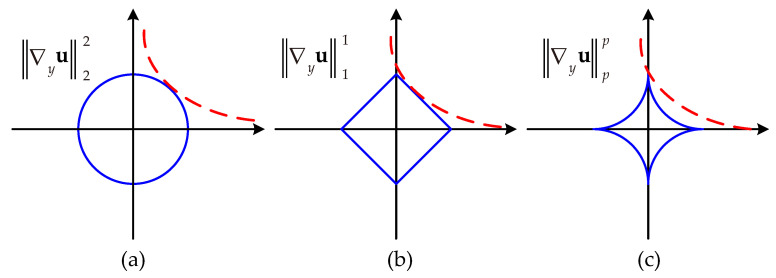
Description of sparsity for various norms. (**a**) p=2. (**b**) p=1. (**c**) 0<p<1.

**Figure 4 sensors-23-07529-f004:**
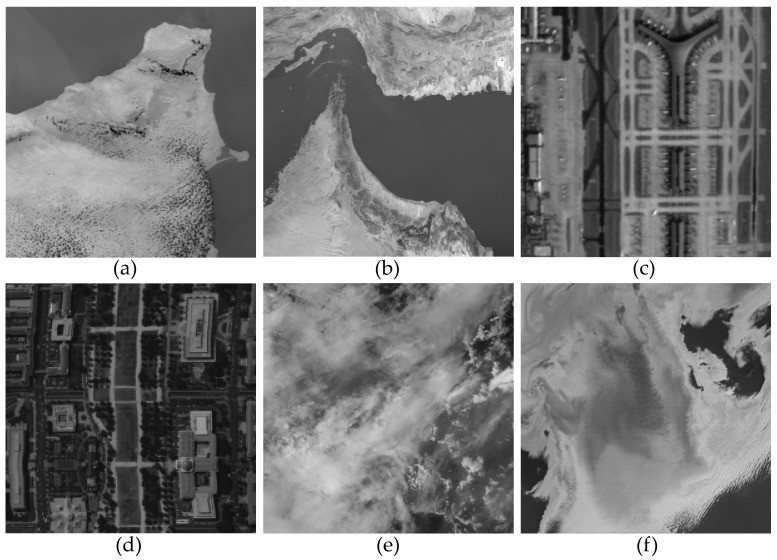
Original images for simulated experiments. (**a**) MODIS band 31 data D1. (**b**) MODIS band 20 data D2. (**c**) Tiangong-1 satellite hyperspectral data D3. (**d**) Washington DC Mall hyperspectral data D4. (**e**) VIIRS hyperspectral data D5. (**f**) MODIS band 31 data D6.

**Figure 5 sensors-23-07529-f005:**
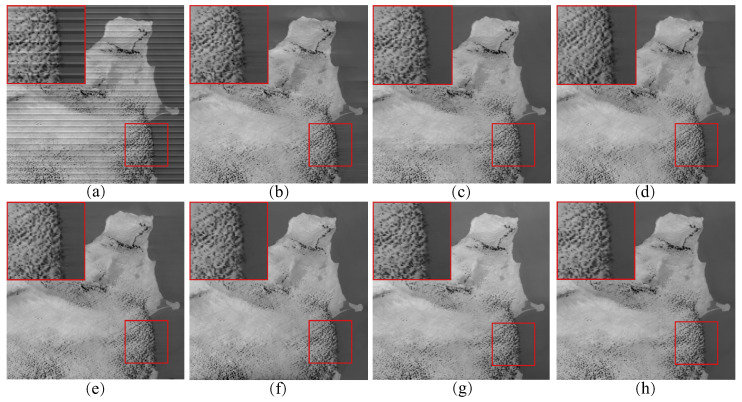
The destriping results of simulated periodic stripe noise in MODIS data band 31 D1. (**a**) Degraded image; (**b**) WAFT; (**c**) WLS; (**d**) UTV; (**e**) SUTV; (**f**) GSLV; (**g**) RBSUTV; (**h**) The proposed method.

**Figure 6 sensors-23-07529-f006:**
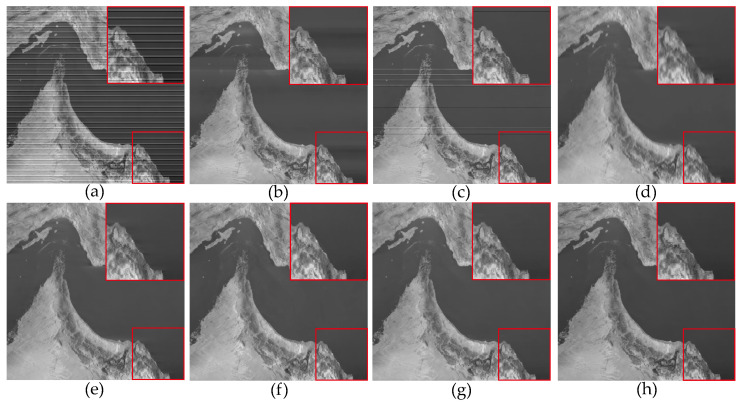
The destriping results of simulated periodic stripe noise in MODIS data band 20 D2. (**a**) Degraded image; (**b**) WAFT; (**c**) WLS; (**d**) UTV; (**e**) SUTV; (**f**) GSLV; (**g**) RBSUTV; (**h**) The proposed method.

**Figure 7 sensors-23-07529-f007:**
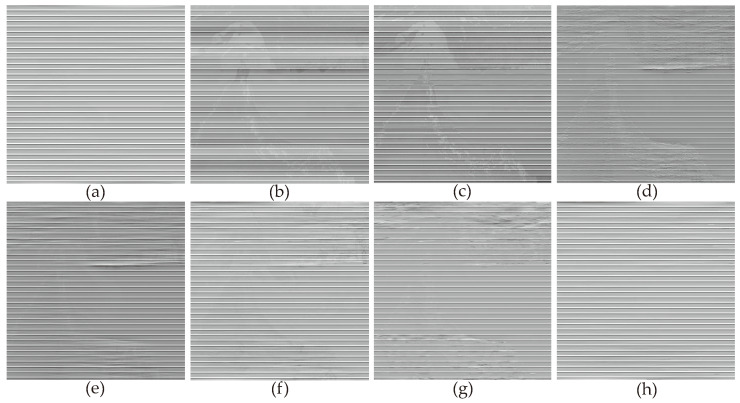
Noise estimation comparison results of Figure 6 images. (**a**) Added stripe noise; (**b**) WAFT; (**c**) WLS; (**d**) UTV; (**e**) SUTV; (**f**) GSLV; (**g**) RBSUTV; (**h**) The proposed method.

**Figure 8 sensors-23-07529-f008:**
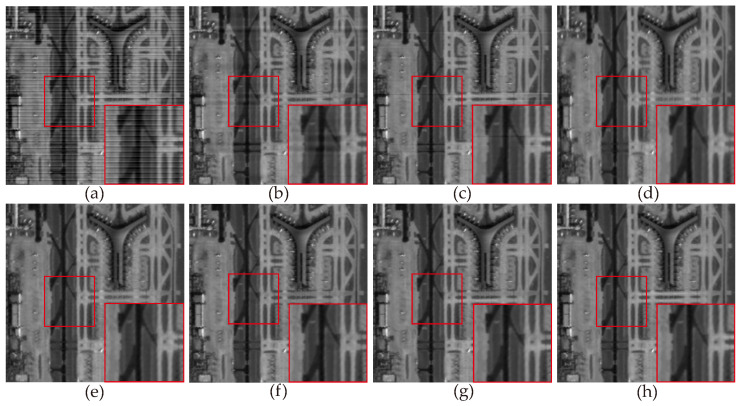
The destriping results of simulated periodic stripe noise in Tiangong-1 satellite hyperspectral data D3. (**a**) Degraded image; (**b**) WAFT; (**c**) WLS; (**d**) UTV; (**e**) SUTV; (**f**) GSLV; (**g**) RBSUTV; (**h**) The proposed method.

**Figure 9 sensors-23-07529-f009:**
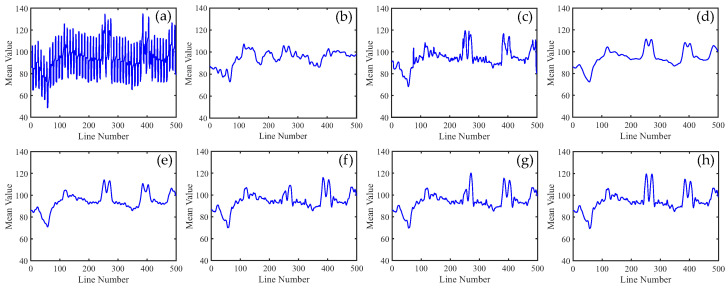
Mean cross-track profiles of Figure 8 images. (**a**) Degraded image; (**b**) WAFT; (**c**) WLS; (**d**) UTV; (**e**) SUTV; (**f**) GSLV; (**g**) RBSUTV; and (**h**) The proposed method.

**Figure 10 sensors-23-07529-f010:**
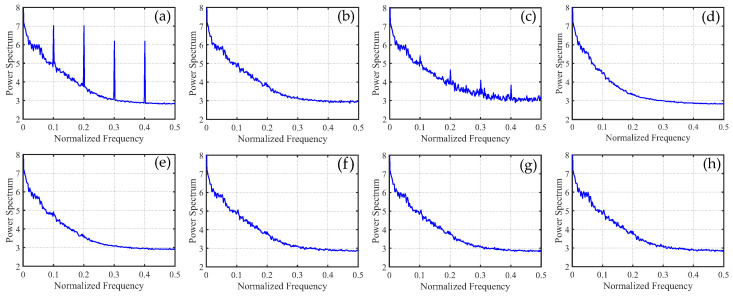
Mean column power spectrum of Figure 8 images. (**a**) Degraded image; (**b**) WAFT; (**c**) WLS; (**d**) UTV; (**e**) SUTV; (**f**) GSLV; (**g**) RBSUTV; and (**h**) The proposed method.

**Figure 11 sensors-23-07529-f011:**
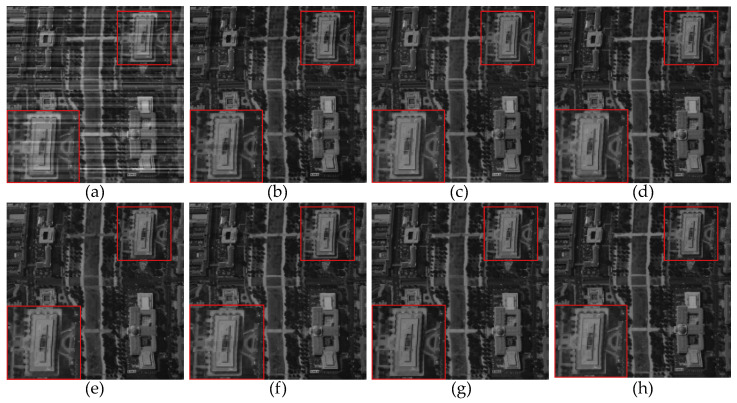
The destriping results of simulated nonperiodic stripe noise in Washington DC Mall hyperspectral data D4. (**a**) Degraded image; (**b**) WAFT; (**c**) WLS; (**d**) UTV; (**e**) SUTV; (**f**) GSLV; (**g**) RBSUTV; (**h**) The proposed method.

**Figure 12 sensors-23-07529-f012:**
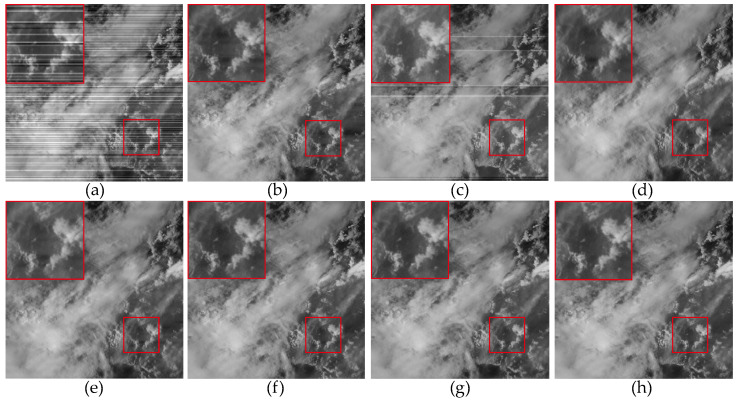
The destriping results of simulated nonperiodic stripe noise in VIIRS hyperspectral data D5. (**a**) Degraded image; (**b**) WAFT; (**c**) WLS; (**d**) UTV; (**e**) SUTV; (**f**) GSLV; (**g**) RBSUTV; (**h**) The proposed method.

**Figure 13 sensors-23-07529-f013:**
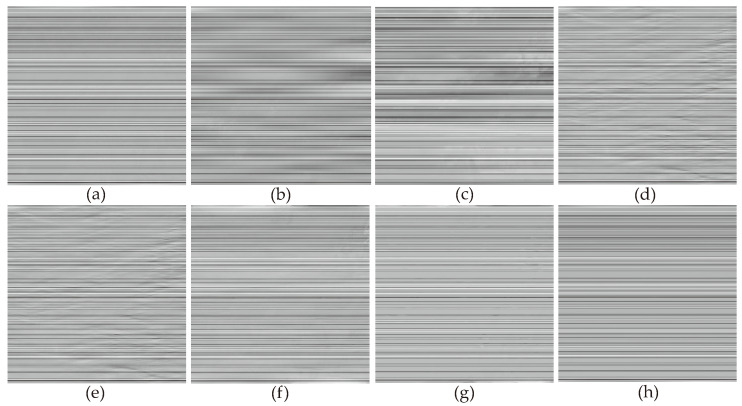
Noise estimation comparison results of Figure 12 images. (**a**) Added stripe noise; (**b**) WAFT; (**c**) WLS; (**d**) UTV; (**e**) SUTV; (**f**) GSLV; (**g**) RBSUTV; (**h**) The proposed method.

**Figure 14 sensors-23-07529-f014:**
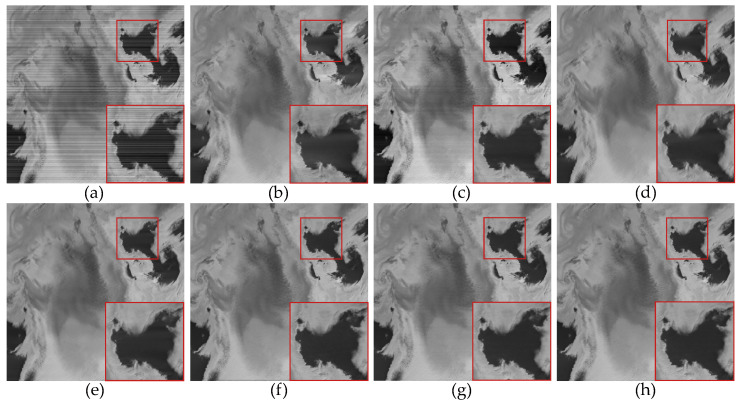
The destriping results of simulated nonperiodic stripe noise in MODIS data band 22 D6. (**a**) Degraded image; (**b**) WAFT; (**c**) WLS; (**d**) UTV; (**e**) SUTV; (**f**) GSLV; (**g**)RBSUTV; (**h**) The proposed method.

**Figure 15 sensors-23-07529-f015:**
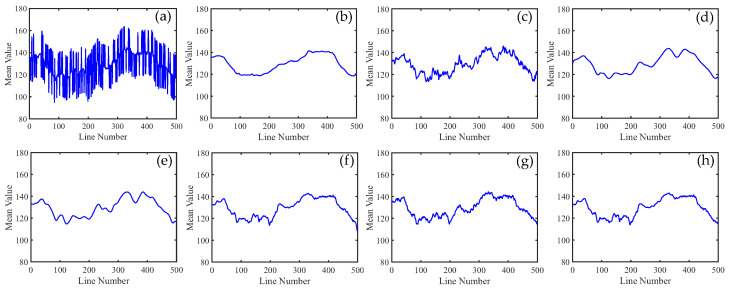
Mean cross-track profiles of Figure 14 images. (**a**) Degraded image; (**b**) WAFT; (**c**) WLS; (**d**) UTV; (**e**) SUTV; (**f**) GSLV; (**g**) RBSUTV; (**h**) The proposed method.

**Figure 16 sensors-23-07529-f016:**
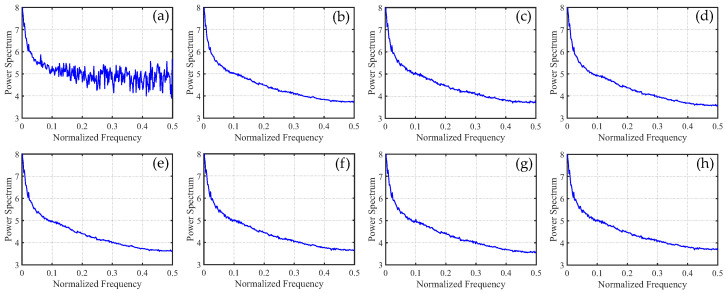
Mean column power spectrum of Figure 14 images. (**a**) Degraded image; (**b**) WAFT; (**c**) WLS; (**d**) UTV; (**e**) SUTV; (**f**) GSLV; (**g**) RBSUTV; (**h**) The proposed method.

**Figure 17 sensors-23-07529-f017:**
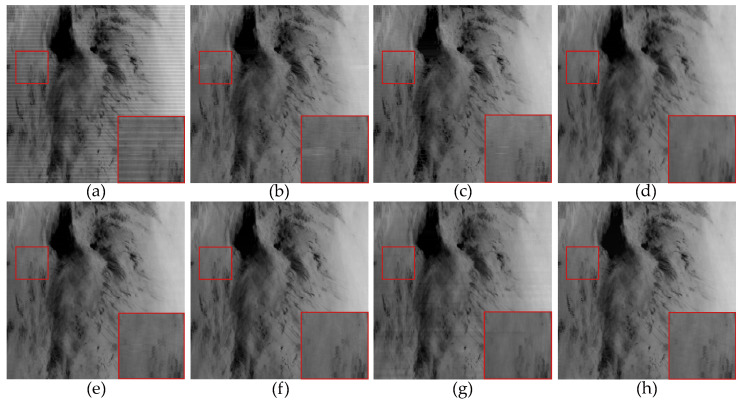
The destriping results of different method in MODIS data band 27 D7. (**a**) Original image; (**b**) WAFT; (**c**) WLS; (**d**) UTV; (**e**) SUTV; (**f**) GSLV; (**g**) RBSUTV; (**h**) The proposed method.

**Figure 18 sensors-23-07529-f018:**
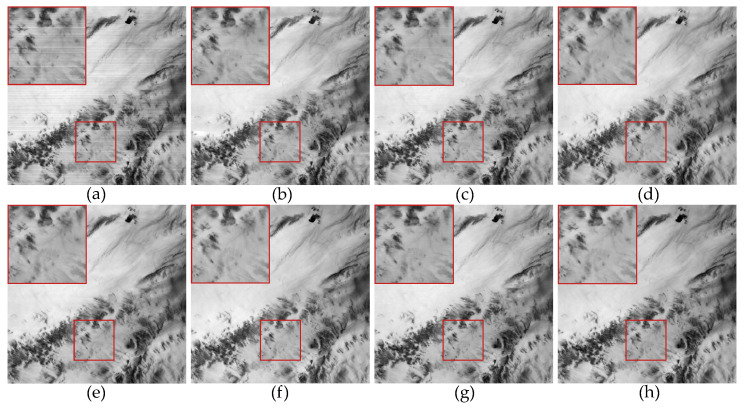
The destriping results of different method in MODIS data band 27 D8. (**a**) Original image; (**b**) WAFT; (**c**) WLS; (**d**) UTV; (**e**) SUTV; (**f**) GSLV; (**g**) RBSUTV; (**h**) The proposed method.

**Figure 19 sensors-23-07529-f019:**
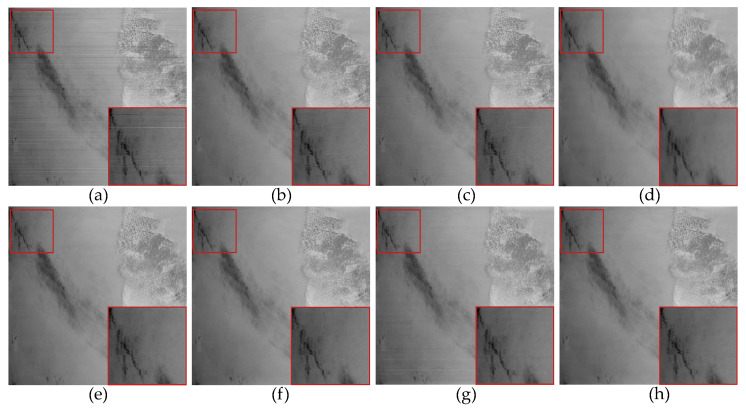
The destriping results of different methods in MODIS data band 33 D9. (**a**) Original image; (**b**) WAFT; (**c**) WLS; (**d**) UTV; (**e**) SUTV; (**f**) GSLV; (**g**) RBSUTV; (**h**) The proposed method.

**Figure 20 sensors-23-07529-f020:**
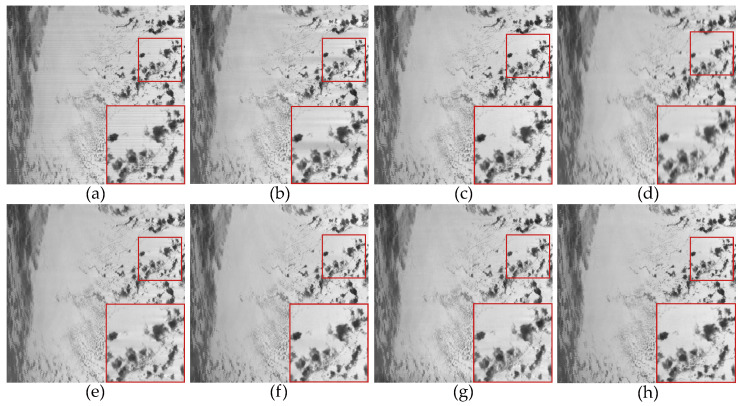
The destriping results of different method in MODIS data band 30 D10. (**a**) Original image; (**b**) WAFT; (**c**) WLS; (**d**) UTV; (**e**) SUTV; (**f**) GSLV; (**g**) RBSUTV; (**h**) The proposed method.

**Figure 21 sensors-23-07529-f021:**
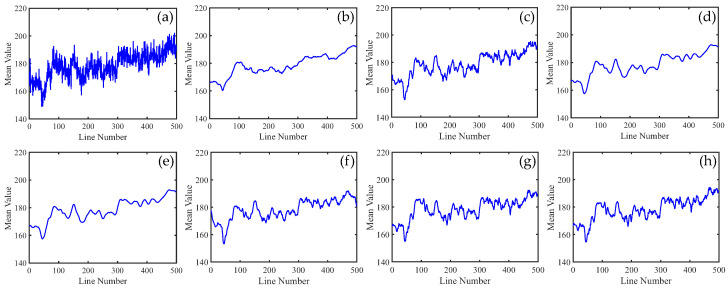
Mean cross-track profiles of Figure 20 images. (**a**) Original image; (**b**) WAFT; (**c**) WLS; (**d**) UTV; (**e**) SUTV; (**f**) GSLV; (**g**) RBSUTV; (**h**) The proposed method.

**Figure 22 sensors-23-07529-f022:**
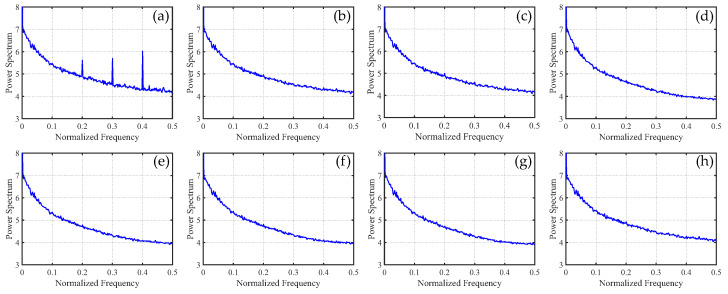
Mean column power spectrum of Figure 20 images.(**a**) Original image; (**b**) WAFT; (**c**) WLS; (**d**) UTV; (**e**) SUTV; (**f**) GSLV; (**g**) RBSUTV; (**h**) The proposed method.

**Figure 23 sensors-23-07529-f023:**
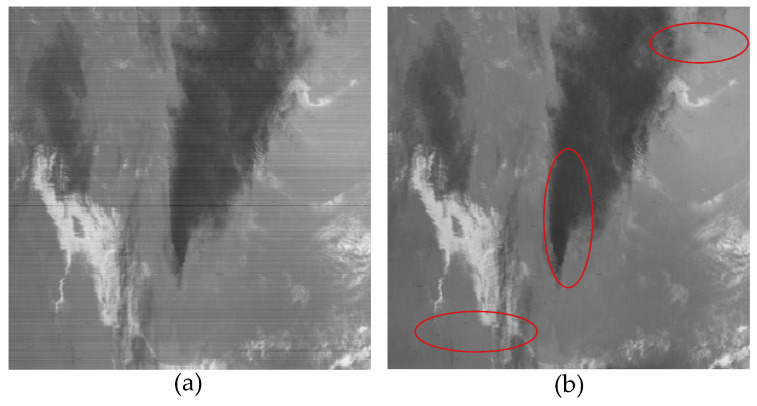
An example of the proposed model failure to completely remove stripes. (**a**) Original image. (**b**) Result of the proposed method.

**Table 1 sensors-23-07529-t001:** The PSNR and SSIM results of MODIS data band 20 D2 with periodic noise.

Image	Method	r = 0.2			r = 0.5			r = 0.8		
		Intensity			Intensity			Intensity		
		20	50	80	20	50	80	20	50	80
PSNR	WAFT	43.4022	41.0526	36.4039	38.7461	34.4116	31.1280	36.7882	33.7920	26.0641
	WLS	46.0724	39.4454	35.3322	41.6714	36.1614	30.6251	35.0809	31.1331	27.3167
	UTV	39.2602	38.1358	34.5581	38.3960	34.8853	32.3572	37.7028	33.2023	30.5506
	SAUTV	41.0473	38.9131	37.4996	39.3513	35.3723	34.7114	36.0768	35.1770	29.5822
	GSLV	43.1521	40.6223	38.0372	41.2094	38.5517	33.7158	39.8349	37.9538	30.5086
	RBSUTV	**48.6146**	**43.7358**	37.4996	41.4725	35.2497	32.3088	33.3342	29.8573	27.9977
	Proposed	46.5692	43.3382	**39.6116**	**44.8139**	**41.0586**	**35.5250**	**41.2979**	**38.6075**	**32.6568**
SSIM	WAFT	0.9920	0.9909	0.9809	0.9853	0.9676	0.9351	0.9792	0.9654	0.9361
	WLS	0.9968	0.9956	0.9922	0.9976	0.9958	0.9881	0.9956	0.9914	0.9309
	UTV	0.9901	0.9878	0.9755	0.9883	0.9774	0.9661	0.9864	0.9697	0.9497
	SAUTV	0.9982	0.9925	0.9918	0.9949	0.9905	0.9885	0.9902	0.9879	0.9585
	GSLV	0.9984	0.9945	0.9932	0.9968	0.9943	0.9852	0.9974	0.9936	0.9644
	RBSUTV	**0.9995**	**0.9986**	0.9882	0.9966	0.9787	0.9742	0.9696	0.9381	0.9155
	Proposed	0.9993	0.9983	**0.9949**	**0.9987**	**0.9982**	**0.9936**	**0.9975**	**0.9946**	**0.9741**

**Table 2 sensors-23-07529-t002:** The PSNR and SSIM results of VIIRS hyperspectral data D5 with non-periodic noise.

Image	Method	r = 0.2			r = 0.5			r = 0.8		
		Intensity			Intensity			Intensity		
		20	50	80	20	50	80	20	50	80
PSNR	WAFT	34.2481	31.4194	30.6256	33.2016	30.7178	29.0722	32.7578	27.5066	24.6842
	WLS	39.3885	33.8543	29.9705	38.9246	32.4222	28.1826	36.4057	26.0033	24.2943
	UTV	36.3369	32.9826	31.8258	33.7618	30.4517	29.9246	32.2136	28.7507	25.9453
	SAUTV	38.2431	33.1458	32.0741	35.7210	32.6039	30.5993	33.7555	30.0501	27.2049
	GSLV	41.0957	40.5213	36.9920	39.2048	36.6313	32.6001	38.8471	34.5522	28.9274
	RBSUTV	**43.0943**	38.4611	35.2614	38.8061	36.9439	32.7291	33.4630	27.5817	24.9959
	Proposed	42.4628	**40.6482**	**37.5273**	**42.0398**	**37.6390**	**33.5871**	**41.0827**	**35.0869**	**30.6429**
SSIM	WAFT	0.9882	0.9813	0.9660	0.9863	0.9679	0.9049	0.9858	0.9243	0.8919
	WLS	0.9974	0.9794	0.9448	0.9950	0.9635	0.9290	0.9921	0.9245	0.8717
	UTV	0.9758	0.9617	0.9551	0.9628	0.9471	0.9339	0.9567	0.9388	0.9004
	SAUTV	0.9885	0.9753	0.9629	0.9803	0.9723	0.9606	0.9774	0.9616	0.9246
	GSLV	0.9987	0.9963	0.9798	0.9982	0.9855	0.9699	0.9968	0.9686	0.9102
	RBSUTV	**0.9997**	0.9952	0.9756	0.9984	**0.9911**	0.9722	0.9801	0.9028	0.8692
	Proposed	0.9995	**0.9967**	**0.9862**	**0.9991**	0.9903	**0.9784**	**0.9976**	**0.9785**	**0.9539**

**Table 3 sensors-23-07529-t003:** The MRD and ICV results of the different methods on real data.

Image	Index	WAFT	WLS	UTV	SAUTV	GSLV	RBSUTV	Proposed
MODIS	MRD (%)	3.9013	5.7286	4.3230	3.9782	**2.9361**	6.9148	3.3134
data D7	ICV	48.49	49.94	63.37	54.09	**73.11**	41.17	56.10
MODIS	MRD (%)	3.6293	1.6754	2.8951	1.2567	3.7674	2.0363	**1.1608**
data D8	ICV	83.97	73.80	80.15	88.23	74.29	84.12	**89.74**
MODIS	MRD (%)	2.3294	1.4604	2.0018	1.1688	1.8633	2.2361	**1.0299**
data D9	ICV	79.15	72.47	107.47	**112.93**	90.29	76.21	101.63
MODIS	MRD (%)	6.2981	5.1454	7.1068	6.2015	4.8334	5.5062	**2.3385**
data D10	ICV	101.13	109.64	163.37	127.50	137.58	114.04	**172.35**

**Table 4 sensors-23-07529-t004:** Computational cost of solving the proposed model with ADMM and Fast ADMM.

Image Size	200 × 200	300 × 300	400 × 400	500 × 500	600 × 600	700 × 700
ADMM	1.1871	2.9263	6.0689	9.2549	13.3545	18.2670
Fast ADMM	0.3352	0.9538	1.8967	3.0324	4.1232	5.8862

## Data Availability

Not applicable.

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
