# Peer review of "Destriping of Remote Sensing Images by an Optimized Variational Model"

_sensors, 2023, doi:10.3390/s23177529_

Round 1
Reviewer 1 Report
The article is well organized. Starting from the sparsity of stripe noise, the application of the total variation method in removing stripe noise is first discussed. Then, quasi norm is introduced to have better global sparsity characteristics. Then, based on the gradient characteristics of the stripe noise image itself, regularization constraints in different directions are designed using quasi norm for processing.However, there are some areas that need to be revised in the paper:
1、There are some issues with the layout of the paper, and the labels in Figures 6-22 do not correspond to the text labels。
2、Figure 4e in Line 293 should be Figure 4d。
3、Line324:Why don't add stripe noise of same ratio to the figure5 and figure6?
4、Is it better to use the information from the original image as the truth value comparison in Figures 9 and 10? the same question to Figure 15 and 16. the same question to Figure 21 and 22.
There are no major issues with the English writing of this paper.
Reviewer 2 Report
Dear Authors:
In the paper, the introduction of the Lp quasinorm to characterize the sparsity of images and stripe noise is quite innovative. The extensive comparative experimental results also indicate that this method is more effective in removing stripe noise. Additionally, the utilization of the fast ADMM to reduce the algorithm's runtime is commendable. In summary, the proposed method in the paper proves to be valuable for stripe noise removal and exhibits a certain level of originality. Therefore, I consider the paper suitable for publication. However, there are still some areas for improvement in the paper:
(1) The paper does not provide a clear enough explanation of the relevant prior characteristics and the properties of the L0 norm used in the proposed model.
(2) It's advisable to avoid using adverbial clauses at the beginning of sentences, as they can reduce the readability of the paper.
(3) You need to pay attention to the correct use of verb tenses in the paper to ensuring a more logical grammatical structure.
(4) The sizing design of certain figures is not well-considered, resulting in a less organized appearance in the layout of the paper.
The Quality of English Language is good.
Reviewer 3 Report
In this manuscript, the author proposed a new unidirectional variational model to remove horizontal stripe noise. The proposed model fully considered the directional characteristics and structural sparsity of the stripe noise, as well as the prior features of the underlying image, to design different sparse constraints. A fast ADMM algorithm was employed in the model-solving process to speed up the convergence during iterations. However, a major revision has to be done before this manuscript can be accepted for publication.
1) Please describe the main differences between this manuscript and the referenced work. Subsequently, I recommend performing a comparative experiment within the experimental section to gauge the improvements introduced by your proposed method accurately. (For instance, it would be beneficial to evaluate the performance of your method in comparison to the referenced approach, utilizing an equivalent ADMM optimizer configuration.). Furthermore, I noticed that the author made improvements and optimizations inspired by the references. Therefore, I recommend that the authors cite this reference.
[1] Liu, Li, Luping Xu, and Houzhang Fang. Simultaneous intensity bias estimation and stripe noise removal in infrared images using the global and local sparsity constraints. IEEE Transactions on Geoscience and Remote Sensing 58.3 (2019): 1777-1789.
2) Please give more detailed experimental proof that the Lp quasinorm better captures the relevant sparsity properties in simulated and real data.
3) In the introduction section, please give the specific limitations of model optimization and the basis for how you can further solve the limitations.
4) In the experiment, please give a specific operating efficiency comparison table so we can see the gap more accurately.
5) Table 1 and Table 2 typesetting needs to be further adjusted to achieve better display results.
Some minor language errors need to be corrected, e.g., an extra period on line 203, an extra colon on line 231, and an extra symbol (2) on line 235.
